# A New Phosphorous/Nitrogen-Containing Flame-Retardant Film with High Adhesion for Jute Fiber Composites

**DOI:** 10.3390/polym15081920

**Published:** 2023-04-18

**Authors:** Yanli Dou, Zheng Zhong, Jiaming Huang, Aixun Ju, Weiguo Yao, Chunling Zhang, Dongbo Guan

**Affiliations:** The Ministry of Education Key Laboratory of Automotive Material, College of Materials Science and Engineering, Jilin University, Changchun 130025, China

**Keywords:** flame retardant coating, UV-curable, fiber needled felts

## Abstract

In this work, a novel P/N flame-retardant monomer (PDHAA) was synthesized through reacting phenyl dichlorophosphate (PDCP) with N-hydroxyethyl acrylamide (HEAA). The structure of PDHAA was confirmed using Fourier transform infrared (FTIR) spectroscopy and proton nuclear magnetic resonance (NMR) spectroscopy. PDHAA monomer and 2-hydroxyethyl methacrylate phosphate (PM-2) monomer were mixed at different mass ratios, to prepare UV-curable coatings, and then applied to the surface of fiber needled felts (FNFs), to improve their flame retardancy. PM-2 was introduced to reduce the curing time of the flame-retardant coatings and improve the adhesion between the coating and the fiber needled felts (FNFs). The research results indicated that the surface flame-retardant FNFs had a high limiting oxygen index (LOI) and rapidly self-extinguished in a horizontal combustion test and passed a UL-94 V-0 test. At the same time, the CO and CO_2_ emissions were greatly reduced, and the carbon residue rate was increased. In addition, the introduction of the coating improved the mechanical properties of the FNFs. Therefore, this simple and efficient UV-curable surface flame-retardant strategy has broad application prospects in the field of fire protection.

## 1. Introduction

Fiber composites (FC) have been a key research object in environmental protection materials in recent years, due to their high output, low price, and low weight [1,2]. In FCs, hot pressed plates made of fiber needled felts (FNFs) can be used with leather and textiles, which has been widely used in the field of automobile interior decoration, household appliances, building decoration, and so on [3]. However, an disadvantage of this environmentally friendly material is also obvious; that is, FNFs are highly flammable, which can lead to fires, causing a great threat to people’s lives and property, and seriously limiting their application in industry [4]. Furthermore, needle felts are made by mixing polymer fibers and natural fibers, and it is difficult to introduce a flame retardant into this mixture of fibers. As a result, a surface flame retardant strategy is a suitable choice for these FNFs.

The surface flame retardant methods for FNFs can be summarized as powder spraying methods and coating methods. Ammonium polyphosphate/expandable graphite (APP/EG) powder was sprayed on the surface of FNFs using a surface powder spraying method, to prepare FNFs with flame retardant properties [5]. The LOI value of the composite was more than 30%, and the mechanical properties were greatly improved. However, it is difficult to evenly distribute the flame retardant on the surface of FNFs with the surface powder spreading method. UV-curable coating technology is a relatively fast and effective method for preparing thin coatings on different types of substrate [6]. The combination of UV-curing technology and a flame-retardant coating to improve the fire resistance performance of the substrate has become an ideal choice [7,8,9,10].

Phosphorus−nitrogen (P−N) flame retardant is a kind of highly efficient halogen-free flame retardant. Due to its low toxicity, low smoke, and lack of corrosive gas during combustion, this kind of flame retardant has been widely used in flame-retardant coatings and polymers [11,12,13,14,15]. At present, P–N flame retardants can be divided into P–N mixed intumescent flame retardants and P–N single molecule intumescent flame retardants. Mixed intumescent flame retardants are realized by compounding and adjusting the ratio of the different phosphorus and nitrogen flame retardants, which are mainly used in epoxy resin [16], polyolefin [17], intumescent coatings [10], etc. Single molecule intumescent flame retardants that integrate a carbon source, acid source, and gas source can show excellent flame-retardant performance and overcome the disadvantages of a low thermal stability and poor compatibility [18,19]. A phosphate–nitrogen-based reactive diluent (DHMPP) was mixed with epoxy acrylate (EA) in certain ratios, to prepare a flame-retardant wood coating. The LOI of the wood coating reached 33% and achieved a UL-94 V-0 rating [20]. The researchers developed a novel flame retardant (ethoxy piperazine (phenyl) phosphoryl ethyl methacrylate (EPPEM), containing phosphorus, nitrogen, and benzene ring, and applied to the surface of PET/CO fabrics. Compared to PET/CO fabrics, the LOI of the UV-cured fabrics containing 300 g/L EPPEM solution increased from 17.8% to 27.2%, indicating that the introduction of EPPEM improved the flame retardancy of the fabric [21].

Phenylphosphonyl chloride contains phosphorus and benzene ring, having a high thermal stability and being a potential acid source. Phenylphosphonyl chloride can react with monomers containing hydroxyl to synthesize phosphorus-containing flame-retardant monomers, which can endow polymers with excellent flame-retardant properties [22,23]. However, there have been few reports on the one-step synthesis of UV-curable monomers with phosphorus, nitrogen, and benzene ring structures. Given the above considerations, we synthesized a novel P–N flame-retardant monomer bis (2-acrylamidoethyl) phenyl phosphate (PDHAA) through a one-step substitution reaction. The synthesis process of the monomers was simple, and the reaction activity was high. PDHAA and PM-2 were mixed at different mass ratios, and then applied as a coating mixture to the surface of FNFs after hot pressing. The structure of the PDHAA monomer was studied using Fourier transform infrared (FTIR) spectroscopy and proton nuclear magnetic resonance (NMR) spectroscopy. The flame retardancy of the FNFs was evaluated using the limiting oxygen index (LOI), vertical combustion (UL-94) test, thermogravimetric analysis (TGA), and cone calorimetry (CCT). In addition, the mechanical properties and adhesion of the FNFs were also tested.

## 2. Experimental

### 2.1. Materials

Phenyl dichlorophosphate (PDCP), N-hydroxyethyl acrylamide (HEAA), and triethylamine (TEA) were provided by Aladdin Reagent Co., Ltd. (Shanghai, China); 2-hydroxyethyl methacrylate phosphate (PM-2) was provided by Guangzhou Lihou Trade Co., Ltd. (Guangzhou, China); and 2-hydroxy-2-methyl-1-phenyl-1-acetone (Darocur 1173) was provided by McLean Co., Ltd. (Shanghai, China); The PDCP, HEAA, and TEA were dried with a 4A molecular sieve before use. Tetrahydrofuran (THF) was provided by Sinopharm Reagent Co., Ltd. (Shanghai, China) and flax/kenaf/polypropylene fiber composites were provided by Bochao Auto Parts Co., Ltd. (Changchun, China).

### 2.2. Preparation of the Phosphorus Nitrogen Flame-Retardant Monomer (PDHAA)

HEAA (2.88 g, 0.025 mol), TEA (3.04, 0.03 mol), and 100 mL THF were added to a 500 mL three-necked flask equipped with a magnetic stirrer. Then, the mixture of PDCP (6.72, 0.05 mol) and 40 mL THF was added to the above reaction flask by dropping at 0 °C and stirred continuously for 12 h. After the reaction was completed, the triethylammonium hydrochloride solid was removed by filtration, followed by distillation under reduced pressure, to remove the solvent and unreacted reactants. The synthesized monomer was dissolved in dichloromethane (DCM) solvent, and the solution was extracted. Finally, the product was further purified using silica gel chromatography (ethyl acetate/methanol = 10:1, *v*/*v*), the yield was 58%. The yellowish liquid product was identified as PDHAA. The synthesis route is given in Figure 1.

### 2.3. Preparation of Phosphorus Nitrogen Flame-Retardant Monomer (PDHAA)

The FNFs were cut to a certain size and pressed under a plate vulcanizing machine at 180 °C and 8 KPa, preheated for 8 min, held for 8 min, removed, and cooled for 10 min, to form the fiber composite plates. The precursor solution was prepared using a simple solution mixing method. The formulations are listed in Table 1. First, PDHAA and PM-2 were mixed with different mass ratios in a beaker under continuous stirring. Next, 4wt% Darocur 1173 as a photoinitiator was added to the above mixture. The whole system was stirred using a glass rod at room temperature for about 20 min. The mixture was then put in a vacuum drying oven for about 30 min at 25 °C, to remove bubbles. Finally, the uniform and yellowish precursor solution was coated on the fiber composite plate with a rod coater, and the coating thickness was 200 μm. It was then exposed to a UV-curing machine with a power of 1 KW and a wavelength of 320–400 nm; the fabrication procedure is displayed in Figure 1.

### 2.4. Characterization

The chemical structure of PDHAA was determined using FTIR spectroscopy (TENSOR 27, Bruker, Saarbruecken, Germany; a wavelength range of 4000 cm^−1^–400 cm^−1^) and nuclear magnetic resonance spectroscopy (NMR; AVANCE-300, Bruker, Saarbruecken, Germany spectrometer at room temperature and the solvent was DMSO-d_6_).

According to GB2406.2-2009 standards, the LOI value of the sample was determined using a JF-3 type oxygen index tester (Festec Company, Seoul, Korea), the dimensions of all specimens were 100 × 10 × 3 mm^3^.

According to the GB 8410-2006 standard, an automotive interior material combustion characteristics horizontal combustion tester (H1011D, Changchun, China) was used to carry out a horizontal combustion test; the dimensions of all specimens were 150 × 75 × 3 mm^3^.

According to the ASTMD3801 standard, the vertical combustion performance of the samples was evaluated with a UL94-X vertical burner (Motis Company, Zongshan, China); the dimension of all specimens were 125 × 13 × 3 mm^3^.

According to the ISO 5660 standard, cone calorimetry was conducted on a cone calorimeter (FTT) at a heat flux of 50 kW/m^2^, with specimen dimensions of 100 × 100 × 3 mm^3^.

A thermogravimetric analysis instrument HTG-1 (Hengjiu Scientific Instrument Factory, Beijing, China) was used to test the samples under nitrogen atmosphere. The weight of each sample was 5–10 mg, the temperature range was 25–700 °C, and the flow rate was 50 mL/min.

SEM scanning images were obtained using a scanning electron microscope JSM-6700F (JEOL Co., Tokyo, Japan).

The Raman spectrometer in Via-Plus532 (Renishaw, Gloucestershire, UK) was used to characterize the composites after horizontal combustion.

A thermogravimetric analysis instrument 209F3 and infrared spectrometer TENSOR27 (Bruker, Saarbruecken, Germany) were used to analyze the volatile gases of the composite. The composite was heated from room temperature to 900 °C at N_2_ flow rates of 20 °C/min.

According to the TL52448 standard, a tensile test was carried out on the samples using a universal testing machine (Intelligent Instrument Equipment, Changchun, China). The sample size was 100 × 25 × 3 mm^3^. The drawing rate was 3 mm/min. The results were the average of three tests.

According to the ISO 2409:2013 standards, adhesion tests were performed using a single-blade grid cutter with a spacing of 3 mm.

## 3. Results and Discussion

### 3.1. Structural Characterization of PDHAA Monomer

PDHAA was synthesized through the reaction between Phenyl dichlorophosphate (PDCP) and N-hydroxyethyl acrylamide (HEAA), and the structure of PDHAA was characterized using FTIR, ^1^H NMR, and ^31^P NMR. Figure 2 shows the FTIR spectrum of PDCP, HEAA, and PDHAA; the peaks at 1668 cm^−1^, 1627 cm^−1^, and 1315 cm^−1^ were ascribed to C=O, C=C, and C-N stretching of the HEAA structure units, respectively [24,25]. The peaks at 1168 cm^−1^, 1026 cm^−1^, and 761 cm^−1^ corresponded to P=O and P-O-C stretching of the PDCP structure units, respectively [26]. All of the above peaks also existed in the spectrum of the PDHAA. In the FTIR spectrum of the PDCP, the peak at 560 cm^−1^ belonged to the characteristic absorption of phosphorus chloride bond, while there were no homologous absorption peaks in the PDHAA spectrum [27]. The resulting P-O-C peak overlapped with the PDCP structure units, indicating that PDHAA was successfully synthesized through the reaction between PDCP and HEAA.

The ^1^H NMR spectrum of PDHAA showed resonances corresponding to all protons of the given structure, as shown in Figure 3, δ_H_ (ppm): 7.17–7.39 (5H, aromatic protons), 5.74–6.48 (6H, -CH=CH) [28], 4.14–4.45 and 3.3–3.5 (8H, P-O-CH_2_-CH_2_), and 8.3–8.5 (2H, -NH). The remaining characteristic peak was the hydrogen proton peak of the solvent ethyl acetate. In the ^31^P NMR spectrum, there was a single peak at δ_p_ = −6.35 ppm, which was the characteristic peak of phosphorus in PDHAA. Therefore, the results of FTIR, ^1^H NMR, and ^31^P NMR were consistent with the expected molecular structure.

### 3.2. Curing Behavior of UV-Curable Coatings

In order to study the UV curing behavior of flame-retardant coatings, the coatings of the FNFs-1 and FNFs-5 samples were selected for real-time FTIR characterization, to study the curing behavior of PM-2 and PDHAA; then C=O was used an as an internal standard peak to calculate the double bond conversion rate of the coating using the area of C=C double bond and C=O peak. As shown in Figure 4a,b, for PM-2 with a single functional group, the peaks at 1637 cm^−1^ and 1722 cm^−1^ corresponded to the C=C and C=O absorption peaks. After 300 s UV irradiation, the C=C absorption peak of the coating decreased; and at this time, the double bond conversion reached 93.4%. As shown in Figure 4c,d, the C=C absorption peak at 1627 cm^−1^ disappeared after 30 s UV irradiation, and the coating had a faster curing rate and double bond conversion rate. This was due to the high double bond density of PDHAA, which increased the initial reaction rate of the coating [29].

### 3.3. Flame Retardancy of the Fiber Composites

The UV-curable coatings determined the flame-retardant properties of the FNFs. Table 1 shows the LOI, HBR, and UL-94 properties of the FNFs and coated FNFs. The FNF samples caught fire quickly, with a low LOI value of 21.6%, and no self-extinguishing was shown in the horizontal and vertical combustion tests. Meanwhile, the LOI value of the FNFs-1 sample reached 26.4%, and it self-extinguished after 65 s in the horizontal combustion test. This was mainly because the phosphate group in PM-2 promoted the formation of a carbon layer during combustion, which slowed down the spread of the flame and improved the flame retardancy of FNFs [30]. However, the FNFs-1 sample did not reach the vertical combustion level, indicating that the flame-retardant effect of the single phosphorus-containing monomer was limited; with the addition of the PDHAA monomer, the LOI of the coated FNFs gradually increased. When PDHAA and PM-2 were blended with a PDHAA/PM-2 weight ratio of 2/1, the LOI of FNFs-4 reached 27.8%, and it achieved rapid self-extinguishing in the horizontal combustion test and passed the UL-94 V-0 test. To contrast the flame retardancy of the FNFs and FNFs-4, the FNF sample served as a control, and vertical combustion testing was conducted with the FNF and FNFs-4 samples, as displayed in Figure 5. Figure 5a shows that the FNF sample quickly ignited and the flame spread quickly, until it completely burnt out [31], indicating that the FNF sample was very flammable. Figure 5b,c shows that the FNFs-4 sample rapidly self-extinguished within 10 s of being ignited, indicating that the flame retardancy of the coated FNFs was obviously improved. For the FNFs-5 sample, the LOI was as high as 28.2%. It self-extinguished rapidly in the horizontal combustion test and reached a V-0 level in the vertical combustion test. The proportion of nitrogen and phosphorus in the UV-curable coating allowed a cooperative effect, further improving the flame retardancy of the FNFs [32].

Cone calorimetry was used to evaluate the combustion behavior of the composites and to predict the fire intensity in a real fire [33]. The combustion behavior of the FNFs, FNFs -1, FNFs-3, and FNFs -5 was studied using cone calorimetry. The thermal parameters and smoke parameters of these samples are shown in Table 2, and the curves of selected data are shown in Figure 6. In Figure 6a, due to the loose fiber structure of the FNFs, the PHRR of FNFs reached 568.35 kW/m^2^ at 48 s and Av-HRR reached 211.76 kW/m^2^ at 11–346 s after ignition. Differently from the FNFs, the PHRR and Av-HRR of the coated FNFs were reduced, and the combustion was delayed. The coated FNFS formed a dense carbon layer on the surface after ignition, which insulated the heat source and extended the combustion time. Figure 6b shows the THR of the FNFs. The THR of the FNFs was 72.3 MJ/m^2^, while the THR value of the FNFs-5 sample dropped to 71.8 MJ/m^2^, which was lower than the other samples.

The smoke produced in a fire is one of the most important factors and directly leads to death due to asphyxia and/or inhalation of toxic gases [32]. Figure 6c,d and Table 2 clearly show the smoke emission behavior of the FNFs, FNFs-1, FNFs-3, and FNFs-5 after combustion. As seen in Table 2, the Av-COY and Av-CO_2_Y of the FNFs samples after combustion were 0.275 kg/kg and 7.9 kg/kg, respectively. Compared with the FNF sample, the Av-COY and Av-CO_2_Y of the coated FNFs decreased significantly; the FNFs-5 sample had the lowest Av-COY and Av-CO_2_Y, reducing to 0.013 kg/kg and 0.9 kg/kg. As shown in Figure 6c, the first peak of the coated sample appeared in the SPR curve soon after combustion, which was related to the decomposition of the phosphate ester group in the coating [34]. The decomposition of the phosphate ester group increased the formation of the carbon layer, which effectively reduced the Av-COY and AV-CO_2_Y. However, the TSP value of the coated FNFs increased significantly, from 3.3 m^2^ to 6.8 m^2^, which was attributed to the large amount of small particles produced by the coating after combustion. Therefore, the cone calorimetry results showed that the FNFs-5 sample had a better comprehensive flame retardancy.

### 3.4. SEM Analysis of Char Morphology

In order to study the variation in the flame retardancy of the different systems, the morphology of the carbon residues of the FNFs after vertical combustion was observed using a scanning electron microscope (SEM). Figure 7 shows an SEM image of the sample residual carbon, with a magnification of 500 times. Figure 7a shows the carbon residue morphology of the FNFs. The FNF sample formed very loose carbon residues after combustion [35], while all coated FNFSs had obvious carbon layer formation after combustion, which prevented heat transfer between the flame zone and the FNFs matrix. As shown in Figure 7b, the fiber composite coated with PM-2 monomer formed a rough and cracked carbon layer structure after combustion. With the increase in the PDHAA monomer components, the carbon layer gradually became coherent and dense. However, there were a few cracks in the coke surface, as seen in Figure 7c,d, because of the improper proportion of phosphorus and nitrogen, which led to the formation of a poor quality carbon layer with open bubbles [36]. As seen in Figure 7e, many wrinkles appeared on the surface of the carbon layer, which may have been due to the shrinkage of the carbon layer caused by the cooling of the coating after combustion [37]. Compared with the carbon layer of the other samples, the carbon layer of the FNFs-5 was dense and coherent. This also confirmed that the FNFs-5 had better flame retardancy.

### 3.5. Thermal Properties of the Fiber Composites

TGA is an easy and effective way to evaluate the thermal stability of polymers [38]. The TGA and DTG curves of all samples under N_2_ and air atmosphere are shown in Appendix A, and the relative data are given in Appendix A. The FNFs had a higher initial decomposition temperature, and the char residue rate was only 7.70 wt.% under a N_2_ atmosphere. In comparison with the FNFs, all coated FNFs had an earlier initial mass loss and a significant increase in carbon residue rate. The earlier initial decomposition temperature was attributed to the lower stability of O=P-O, C-N bonds in the UV cured coating, which was less stable than the common C–C bond [11,27,39]. With the increase of PDHAA monomer content, the carbon residue rate of the FNFs-4 and FNFs-5 samples decreased, which was due to the increase in the nitrogen content in the coating. For the coated FNFs under air atmosphere, and the coated FNFs continued to decompose above 500 °C. In addition, the initial degradation temperature(T_0.05_), maximum degradation temperature (T_max_), and residual carbon rate of all samples were lower than in the nitrogen atmosphere, which indicated that oxygen accelerated the degradation of the polymers [40].

The DTG curves showed the degradation process of the flame-retardant composites. For pure FNFs, there was a three-step thermal degradation process, corresponding to the thermal decomposition of the hemicelluloses (218–320 °C), cellulose (320–390 °C), and PP (390–500 °C) under N_2_ atmosphere [5]. The coated FNFs had small peaks at 200–300 °C and 300–390 °C, which may have been due to the initial rupture of the P-O-C and C–N bonds. The stage at 420–520 °C was assigned to the breakage of the acrylate main chain and the PP chain in the matrix. In the air atmosphere, the FNFs had a large degradation peak at 400–500 °C, followed by a very low carbon residue rate above 600 °C, which was attributed to the poor thermal stability of the pure FNFs in this temperature range. The coated FNFs degraded slowly above 400 °C but still had a high carbon residue rate at 700 °C. These results indicated that the coating components improved the thermal stability of FNFs and prevented the FNFs from further degradation.

### 3.6. Raman Spectroscopic Characterization of Carbon Residues

In order to explore the degree of graphitization of the carbon layer after combustion of the FNFs and FNFs-5, Raman spectrum tests were conducted on the carbon residue samples of FNFs and FNFs-5. As seen in Figure 8, the ID/IG value of the carbon residue of the FNFs sample was 0.88, while that of the carbon residue of the FNFs-5 sample was 0.84, indicating that the carbon residue of the FNFs-5 sample had a higher degree of graphitization. The carbon layer structure was more stable and effectively inhibited the combustion of the FNFs.

### 3.7. Flame-Retardant Mode of Action of the UV-Curable Coatings

The flame retardant mode of action of PDHAA can be divided into gas phase flame retardant and condensed phase flame retardant processes. The gas phase products of the FNF and FNFs-5 samples were analyzed through TGA-FTIR testing. In Figure 9a it can be seen that the strength of the volatile products in the FNFs-5 sample was significantly lower than that in FNF samples, indicating that the introduction of flame-retardant coatings reduced the generation of volatile products. The characteristic peaks of gas products in Figure 9b,c are similar, including hydrocarbon compounds (2960 cm^−1^), CO_2_ (2354 cm^−1^), carbonyl compounds (1760 cm^−1^), and aromatic compounds (1518 cm^−1^). Unlike the FNF samples, P=O and P-O-C peaks appeared at 1257 cm^−1^, 1184 cm^−1^, and 1030 cm^−1^, as shown in Figure 9c. The absorption peaks completely disappeared at 320 °C, indicating the degradation of P-O-C and the formation of PO• radicals, which eliminated the flammable H• and OH• radicals in the gas-phase system, thereby exerting a quenching effect [41]. When the temperature was further increased to 440 °C, a new peak (3067 cm^−1^) related to volatile aromatic compounds appeared, reflecting the fracturing of the coating and polypropylene chain segments. PDHAA was thermally decomposed to release NH_3_ (931 cm^−1^) when the temperature was increased from 400 °C to 500 °C. The gas phase analysis results of the coated fiber composites indicated that the thermal decomposition of PDHAA released non-combustible gases, such as NH_3_ and CO_2_, which diluted the oxygen concentration around the fiber composites and inhibited the spread of the flame.

In order to study the structure of the carbon residues after the combustion of the coated fiber composites, the carbon residues after combustion of the FNF and FNFs-5 samples was characterized using FTIR, as shown in Figure 10, and a flame-retardant mechanism diagram of the coated fiber composites is displayed in Figure 11. The FTIR analysis of the carbon residues after combustion of the FNF sample showed that the peaks at 1593 cm^−1^ and 1026 cm^−1^ were aromatic carbon and C-O vibration bands. In the carbon residues of the FNFs-5 sample, the peaks at 1260 cm^−1^ and 1090 cm^−1^ corresponded to the POO- groups and P-O vibrational bands [5]. The FTIR results indicated that there were polyphosphoric acid and phosphates in the residual carbon, and the phosphate compounds formed during the combustion process of the coating promoted the formation of the carbon layer and covered the surface of the fiber composite material. The carbon layer could effectively isolate the sample from heat and oxygen. Therefore, the cooperative effect of the phosphorus and nitrogen elements in the coating gave the fiber composites excellent flame retardancy.

### 3.8. Mechanical Properties of the FNFs

Adhesion tests and mechanical property tests of the FNFs and coated FNFs were carried out. It is generally required that the adhesion between a coating and FNFs reaches level 3, which is considered acceptable. Table 3 lists the adhesion and tensile properties of the FNFs and coated FNFs. It can be seen from Table 3 that the adhesion of FNFs-1 reached level 2, and the adhesion of FNFs-5 sample had no level. In addition, with the increase of PDHAA monomer components, the adhesion between the coating and the substrate deteriorated. The reasons for this difference were as follows: as a single functional active diluent, the PM-2 monomer had low viscosity and volume shrinkage, and the phosphate groups in the monomer could penetrate into the shallow surface of PP, improving the adhesion between the coating and the FNFs. With the addition of PDHAA monomer, the content of phosphate ester in the coating was gradually reduced. Meanwhile, the curing of the PDHAA monomer was fast, and the internal stress could not be released, resulting in the large volume shrinkage and high surface tension of the coating, which reduced the adhesion between the coating and FNFs [42,43].

The tensile strength of the FNF sample was 16.60 ± 1.09 MPa, while the tensile strength of the coated FNFs was clearly improved. The tensile strength of the FNFs-1 and FNFs-5 samples increased to 19.16 ± 1.28 MPa and 20.63 ± 1.28 MPa, respectively. The coating containing PDHAA monomer formed a linear polymer after curing, the molecular chain easily became tangled, and the coating had a good toughness after curing. However, due to its poor adhesion, the improvement in the mechanical properties was limited. When PM-2 and PDHAA were blended together with a PM-2/PDHAA weight ratio of 2/1, the adhesion of FNFs-2 reached level 3, and the coating combined well with FNFs-2, dispersing the external force exerted on the composite material. The tensile strength of the FNFs-2 sample reached 26.19 ± 2.35 MPa. Compared with the FNFs, the tensile strength of FNFs-2 was improved by 58%. Therefore, the tensile test results showed that the surface flame retardant strategy had a positive effect on the mechanical properties of the FNFs.

## 4. Conclusions

A novel P/N flame-retardant monomer (PDHAA) was successfully synthesized. The coating mixture was prepared by mixing PDHAA monomer and PM-2 monomer at different mass ratios, and then the coating mixture was applied to the surface of FNFs, obtaining coated FNFs with flame retardancy. With the increase of the PDHAA monomer content, the LOI value of the sample showed an upward trend. The LOI value of FNFs-5 reached 28.8%, it also self-extinguished rapidly in the horizontal combustion test and passed the UL-94 V-0 test. At the same time, the coating changed into a smooth and dense carbon layer and released non-flammable gases during combustion, such as CO_2_ and NH_3_. The formation of the carbon layer reduced the heat transfer between the flame and the substrate, also greatly reducing the emissions of CO and CO_2_. The FNFs-5 sample had the best flame retardancy. In addition, the introduction of PDHAA improved the mechanical properties of the coating. When PM-2 and PDHAA were blended together with a PM-2/PDHAA weight ratio of 2/1, the tensile strength of FNFs-2 was increased by 58% compared to FNFs. However, with a further increase in the PDHAA content, the adhesion between the coating and FNFs deteriorated, and FNFs-1 had the best adhesion. In summary, FNFs with PDHAA monomers had a high flame retardant efficiency, and this is of great significance for developing UV-curable coatings containing multiple flame retardant elements. These coated FNFs have wide application prospects in the field of fire protection for car interior decorations.

## Data Availability

The data presented in this study are available upon request from the corresponding author.

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
