# Peer review of "A New Phosphorous/Nitrogen-Containing Flame-Retardant Film with High Adhesion for Jute Fiber Composites"

_polymers, 2023, doi:10.3390/polym15081920_

Round 1

Reviewer 1 Report

1. There are too many flame retardants with similar structure to PDHAA. The author should further explain the innovation of the work. Compare the flame retarding effect of flame retardants with similar structures, reflecting the progressiveness of this work.

2. The research on the flame retardant mechanism is too simple, and has not systematically studied the gas and condensation mechanism of the flame retardant. The flame retardant mechanism needs to be further characterized by TG-IR, Raman, XPS, SEM and other means.

3. The author emphasized the high adhesion in the title, but did not explain its mechanism, nor did it compare with the adhesion performance of relevant studies.

Author Response

Reviewer #1:

  1. There are too many flame retardants with similar structure to PDHAA. The author should further explain the innovation of the work. Compare the flame retarding effect of flame retardants with similar structures, reflecting the progressiveness of this work.
  2. The research on the flame retardant mechanism is too simple, and has not systematically studied the gas and condensation mechanism of the flame retardant. The flame retardant mechanism needs to be further characterized by TG-IR, Raman, XPS, SEM and other means.
  3. The author emphasized the high adhesion in the title, but did not explain its mechanism, nor did it compare with the adhesion performance of relevant studies.

Response: Thank you for pointing out this mistake, which we have added relevant tests. The relevant contents were highlighted in the revised manuscript. The revised parts are as follows:

  1. Page 2, Section 1 Introduction, “Give the above considerations, it is of great significance to synthesize a single molecule flame retardant with acid source and gas source, we have synthesized a novel P-N flame-retardant monomer (PDHAA).” was revised as “However, there are few reports on the one-step synthesis of UV-curable monomers with phosphorus, nitrogen, and benzene ring structures. Give the above considerations, we have synthesized a novel P-N flame-retardant monomer bis (2-acrylamidoethyl) phenyl phosphate (PDHAA) through a one-step substitution reaction. The synthesis process of the monomers is simple, and the reaction activity is high.”

  1. Page 11,12 Supplementary data: Raman spectra and TG-FTIR

3.6 Raman spectroscopic characterization of carbon residues

In order to explore the graphitization degree of the carbon layer after combustion of FNFs and FNFs-5. Raman spectrum tests were conducted on the carbon residue samples of FNFs and FNFs-5. In Figure.8, the ID/IG value of the carbon residue of FNFs sample is 0.88, while that of the carbon residue of FNFs-5 sample is 0.84, indicating that the carbon residue of FNFs-5 sample has a higher degree of graphitization. The carbon layer structure is more stable and can effectively inhibit the combustion of FNFs.

The flame retardant mechanism of PDHAA can be divided into gas phase flame retardant and condensed phase flame retardant. The gas phase products of samples FNFs and FNFs-5 were analyzed through TGA-FTIR testing. From Figure.9 (a), it can be seen that the strength of volatile products in sample FNFs-5 is significantly lower than that in sample FNFs, indicating that the introduction of flame retardant coatings reduces the generation of volatile products. The characteristic peaks of gas products in Figures.9 (b) and (c) are similar, including hydrocarbon compounds (2960 cm-1), CO2 (2354 cm-1), carbonyl compounds (1760 cm-1), and aromatic compounds (1518 cm-1). Different from FNFs samples, P=O and P-O-C peaks appear at 1257cm-1, 1184cm-1 and 1030cm-1 in Figure.9(c), and the absorption peaks at 1257cm-1, 1184cm-1 and 1030 cm-1 decrease rapidly with the increase of temperature. The absorption peaks disappear completely at 420℃ and 320℃ respectively, indicating that the phosphate groups in PDHAA monomer degrade and form gaseous phosphorus compounds. When the temperature further increased to 440℃, a new peak (3067 cm-1) related to volatile aromatic compounds appeared, reflecting the fracture of the coating and polypropylene chain segments. PDHAA is thermally decomposed to release NH3 (931 cm-1) when the temperature rises from 400℃ to 500℃. The gas phase analysis results of coated fiber composites indicate that the thermal decomposition of PDHAA releases non-combustible gases such as NH3 and CO2, which can dilute the oxygen concentration around the fiber composites and inhibit the spread of flame.

  1. Page 13, Section 3.8 Thermal properties, “The reasons for this difference are as follows: as a single functional active diluent, PM-2 monomer has low curing shrinkage and can improve the adhesion between the coating and FNFs, while the curing speed of PDHAA monomer coating is fast, and the internal stress cannot be released, resulting in large volume shrinkage and high surface tension of the coating, which reduces the adhesion between the coating and FNFs.” was revised as “The reasons for this difference are as follows: as a single functional active diluent, PM-2 monomer has low viscosity and volume shrinkage, and the phosphate groups in the monomer can penetrate into the shallow surface of PP, improving the adhesion between the coating and the FNFs. With the addition of PDHAA monomer, the content of phosphate ester in coating is gradually reduced. Meanwhile, the curing speed of PDHAA monomer is fast, and the internal stress cannot be released, resulting in large volume shrinkage and high surface tension of the coating, which reduces the adhesion between the coating and FNFs.”

Reviewer 2 Report

Manuscript can be published in "Polymers" after minor revision according to the comments given below.

1. Abbreviation of PDHAA monomer should be explained. 

2. Abbreviations in Introduction (DHMPP, EA, CCT etc.) should be explained in detail.

3. Dimensions of specimens for LOI study should be checked and corrected.

4. Figure 2b is hardly readable.

5. Values given in the Table 1 ("...26.4%...") and in the text on the page 5 ("...26.5%...") should be equal each other. 

Author Response

  1. Abbreviation of PDHAA monomer should be explained.
  2. Abbreviations in Introduction (DHMPP, EA, CCT etc.) should be explained in detail.
  3. Dimensions of specimens for LOI study should be checked and corrected.
  4. Figure 2b is hardly readable.
  5. Values given in the Table 1 ("...26.4%...") and in the text on the page 5 ("...26.5%...") should be equal each other.

Response: Thank you for bringing up this important question which we ignored. The spelling was corrected and relevant contents were highlighted in the revised manuscript. The point-by-point responses to the comments are as follows:

  1. Page 2, Section 1 Introduction, “we have synthesized a novel P-N flame-retardant monomer PDHAA” was revise as “we have synthesized a novel P-N flame-retardant monomer bis (2-acrylamidoethyl) phenyl phosphate (PDHAA) through a one-step substitution reaction”.
  2. Page 2, Section 1 Introduction, DHMPP, EA, CCT were revise as“ phosphate–nitrogen-based reactive diluent (DHMPP), epoxy acrylate (EA), cone calorimetry (CCT).
  3. Page 4, Section 2.5. Characterization, “Dimension of specimens for LOI was revise as 100×10×3 mm3.”
  4. 2b was revise as Figure.3, the explanation is as follows:

The 1H NMR spectrum of PDHAA shows the resonance corresponding to all protons of a given structure , as shown in Figure.3, δH (ppm): 7.17-7.39 (5H, aromatic protons), 5.74-6.48 (6H, - CH=CH), 4.14-4.45 and 3.3-3.5 (8H, P-O-CH2-CH2), 8.3-8.5 (2H, -NH). The remaining characteristic peak is the hydrogen proton peak of the solvent ethyl acetate. In the 31P NMR spectrum, there is a single peak at d=-6.35 ppm, which is the characteristic peak of phosphorus in PDHAA. Therefore, the results of FTIR, 1H NMR and 1P NMR are consistent with the expected molecular structure.

  1. Values given in the Table 1 ("...26.4%...") and in the text on the page 5 ("...26.4%...") were equal each other.

Reviewer 3 Report

Notes:

Abstract

The name accepted in the literature is proton NMR spectroscopy, not hydrogen spectroscopy.

Introduction

It should be briefly indicated what flame retardants are generally used to produce fire-resistant non-woven materials? It is necessary to justify why phosphorus-nitrogen flame retardants have advantages over other flame retardants. As an argument, consider phosphazenes, which very effectively reduce the flammability of polymers (for example, https://doi.org/10.3390/polym14173592; https://doi.org/10.3390/polym14245334; https://doi.org/10.1134/S2075113319060273 ).

What is the LOI value of PET fabrics treated with flame retardant and without flame retardant (lines 60,61)?

What amount of flame retardant was applied to the surface of the PET/CO fabric in work 18?

Give the full name of the flame retardant APP/EG (line 38).

Decipher all abbreviations given in the text, for example, give the name of the flame retardant EPPEM (line 59).

Experimental

It is necessary to indicate the countries producing the reagents.

You must specify the PDHAA yield. The reaction scheme should show the formation of the triethylammonium chloride salt.

Apparently, d6-DMSO was used as a solvent for NMR spectroscopy, and not DMSO? (line 118)

Results and discussion

Note on the FTIR spectra of the original PDCP and PDHAA the characteristic absorption peaks of the P-Cl bond.

On the NMR spectrum, mark the chemical shift scale as δH. The NMR spectrum contains signals that the authors did not characterize in the region of 6.5-7.0 ppm, 1-2.3 ppm. Should specify how the authors purify PDHAA from BHT inhibitor? If it has not been purified, it is necessary to purify the product and perform elemental analysis, provide phosphorus and carbon NMR spectra.

What is the reason for the higher activity of the synthesized phosphate during UV curing compared to methacrylate. Give a detailed explanation (lines 173, 174). In figure 4, it is necessary to give explanations for points a), b), c).

What is the reason for the self-extinguishing of samples with the addition of PDHAA compared to PM-2? How do nitrogen atoms in PDHAA affect the combustion process? It is desirable to transfer the TGA curves shown in Figure 6 to Supplemental materials. In Table 4, the name should be corrected to tensile strength instead of tensile strengtg. What is the reason for the higher tensile strength value for FNF-2 compared to FNF-5? (table 4).

According to ISO 2409:2013 (lattice notch method), adhesion grades of 2,3 and 4 indicate poor adhesion of the coating to the felt due to peeling of the coating. Are these values acceptable for the intended applications?

Conclusions

The conclusions, in fact, duplicate the annotation. This section should be revised. In view of the fact that the discussion of the results is divided into a large number of sections, it is necessary to summarize the results in the conclusion section. How does the ratio of monomers affect one or another parameter, what worsens, what improves? What, in the end, is the optimal composition of the composition?

It is necessary to indicate the prospects for further research of the obtained flame retardant, to suggest directions for the use of coated FNF.

In general, the manuscript makes a positive impression and can be published after revision.

Author Response

Response: Thank you for your valuable comments.The point-by-point responses to the comments are as follows:

  1. The name accepted in the literature is proton NMR spectroscopy, not hydrogen spectroscopy.

Page1, Section Absract “hydrogen spectroscopy” was revise as “proton nuclear magnetic resonance (NMR) spectrum”.

  1. Introduction

It should be briefly indicated what flame retardants are generally used to produce fire-resistant non-woven materials? It is necessary to justify why phosphorus-nitrogen flame retardants have advantages over other flame retardants. As an argument, consider phosphazenes, which very effectively reduce the flammability of polymers (for example, https://doi.org/10.3390/polym14173592;https://doi.org/10.3390/polym14245334;https://doi.org/10.1134/S2075113319060273).

Page2.Section Introduction, reference to relevant literature on phosphazenes, “Phosphorus−nitrogen (P−N) flame retardant is a kind of high efficient halogen-free flame retardant with a bright future. Because of its low toxicity, low smoke and no corrosive gas during combustion, this kind of flame retardant has been widely used in flame retardant coatings.” was revise as “Phosphorus−nitrogen (P−N) flame retardant is a kind of high efficient halogen-free flame retardant with a bright future. Because of its low toxicity, low smoke and no corrosive gas during combustion, this kind of flame retardant has been widely used in flame retardant coatings and polymer”.

What is the LOI value of PET fabrics treated with flame retardant and without flame retardant (lines 60,61)? What amount of flame retardant was applied to the surface of the PET/CO fabric in work 18?

Yang et al. develop a novel flame retardant (ethoxy piperazine (phenyl) phosphoryl ethyl methacrylate (EPPEM) containing phosphorus, nitrogen, and benzene ring, and apply on the surface of PET/CO fabrics. Compared to PET/CO fabrics, the LOI of UV-cured fabrics containing 300 g/L EPPEM solution increased from 17.8% to 27.2%, indicating the introduction of EPPEM improves the flame retardancy of the fabric.

Give the full name of the flame retardant APP/EG (line 38)

“APP/EG” was revise as “ammonium polyphosphate/expandable graphite (APP/EG)”

Decipher all abbreviations given in the text, for example, give the name of the flame retardant EPPEM (line 59).

DHMPP was revise as phosphate–nitrogen-based reactive diluent (DHMPP), EA was revise as epoxy acrylate (EA), EPPEM was revise as (ethoxy piperazine (phenyl) phosphoryl ethyl methacrylate (EPPEM), PDHAA was revise as bis (2-acrylamidoethyl) phenyl phosphate (PDHAA)

  1. Experimental

It is necessary to indicate the countries producing the reagents.

All the production reagents are added to China.

You must specify the PDHAA yield. The reaction scheme should show the formation of the triethylammonium chloride salt.

Page 2, 3 Section 2.2 “After the reaction was completed, the hydrochloride by-products were removed by filtration, and 0.5wt% of HEAA monomer BHT copolymer inhibitor was added to the mixture, followed by distillation under reduced pressure to remove the solvent and unreacted reactants” was revise as “After the reaction was completed, the trimethylamine hydrochloride solid was removed by filtration, followed by distillation under reduced pressure to remove the solvent and unreacted reactants. The synthesized monomer was dissolved in dichloromethane (DCM) solvent and the solution was extracted. Finally, the product was further purified by silica gel chromatography (ethyl acetate/methanol =10:1, v/v), yield was 58%”.

Apparently, d6-DMSO was used as a solvent for NMR spectroscopy, and not DMSO? (line 118)

Page4.Section 2.4 “DMSO” was revise as “DMSO-d6”.

Results and discussion

Note on the FTIR spectra of the original PDCP and PDHAA the characteristic absorption peaks of the P-Cl bond.

Page 5.Section3.1, “In the FTIR spectrum of PDCP, the peak at 583–541cm-1 belongs to characteristic absorption of phosphorus chloride bond” was revise as “In the FTIR spectrum of PDCP, the peak at 560cm-1 belongs to characteristic absorption of phosphorus chloride bond.”

Figure. 2 (a) FTIR spectrum of PDCP, HEAA, PDHAA

On the NMR spectrum, mark the chemical shift scale as δH. The NMR spectrum contains signals that the authors did not characterize in the region of 6.5-7.0 ppm, 1-2.3 ppm. Should specify how the authors purify PDHAA from BHT inhibitor? If it has not been purified, it is necessary to purify the product and perform elemental analysis, provide phosphorus and carbon NMR spectra.

“The 1H NMR spectrum of PDHAA shows the resonance corresponding to all protons of a given structure , as shown in Figure.2(b), d (ppm): 7.17-7.39 (5H,aromatic protons), 5.74-6.48 (6H, - CH=CH), 3.24-4.09 (8H, P-O-CH2-CH2), 7.91-8.77 (2H, -NH), the rest are the hydrogen proton peaks of the BHT inhibitor and solvent. Therefore, the results of FTIR and 1HNMR are consistent with the expected molecular structure.” was revise as “In Figure.3, the 1H NMR spectrum of PDHAA shows the resonance corresponding to all protons of a given structure , as shown in Figure.3, δH (ppm): 7.17-7.39 (5H,aromatic protons), 5.74-6.48 (6H, - CH=CH), 4.14-4.45 and 3.3-3.5 (8H, P-O-CH2-CH2), 8.3-8.5 (2H, -NH). The remaining characteristic peak is the hydrogen proton peak of the solvent ethyl acetate. In the 31P NMR spectrum, there is a single peak at δP =-6.35 ppm, which is the characteristic peak of phosphorus in PDHAA. Therefore, the results of FTIR, 1H NMR and 31P NMR are consistent with the expected molecular structure”.

Figure.3 (a) 1H NMR and 31P NMR spectrum of PDHAA

What is the reason for the higher activity of the synthesized phosphate during UV curing compared to methacrylate. Give a detailed explanation (lines 173, 174). In figure 4, it is necessary to give explanations for points a), b), c).

This is due to the high double bond density of PDHAA, which increases the initial reaction rate of the coating.

Figure.4 was revise as Figure.5“Figure. 4 Vertical combustion behavior of FNFs and FNFs-4 samples” was revise as “Figure. 5 (a) Vertical combustion behavior of FNFs, (b) and (c) vertical combustion behavior of FNFs-4 twice ignited”.

What is the reason for the self-extinguishing of samples with the addition of PDHAA compared to PM-2? How do nitrogen atoms in PDHAA affect the combustion process? It is desirable to transfer the TGA curves shown in Figure 6 to Supplemental materials. In Table 4, the name should be corrected to tensile strength instead of tensile strengtg. What is the reason for the higher tensile strength value for FNF-2 compared to FNF-5? (table 4).

Page7, Section3.3 “This is because the proper proportion of nitrogen and phosphorus in the UV-curable coating can realize the synergistic flame retardant effect, further improve the flame retardancy of FNFs.”

Page12, Section3.7, added TG-FTIR test, “When the temperature rises from 400℃ to 500℃, PDHAA is thermally decomposed to release NH3 (931 cm-1). The gas phase analysis results of coated fiber composites indicate that the thermal decomposition of PDHAA releases non-combustible gases such as NH3 and CO2, which can dilute the oxygen concentration around the fiber composites and inhibit the spread of flame”.

Page10, Section 3.5 TGA curves and relative data were transfer to Supplemental materials.

Page14, Section 3.8, “Table 4” was revise as “Table 3”. In Table 3, “Tensile strengtg” was revise as “Tensile strength”

When PM-2 and PDHAA is blended together with a PM-2/PDHAA weight ratio of 2/1, the adhesion of FNFs-2 can reach level 3, and the coating combines well with FNFs-2, dispersing the external force exerted on the composite material. The tensile strength of FNFs-2 sample can reach 26.19±2.35MPa. Compared with FNFs, the tensile strength of FNFs-2 is improved by 58%.

According to ISO 2409:2013 (lattice notch method), adhesion grades of 2,3 and 4 indicate poor adhesion of the coating to the felt due to peeling of the coating. Are these values acceptable for the intended applications?

Page13, Section3.8, It is generally required that the adhesion between the coating and FNFs reaches level 3, which is considered acceptable.

  1. Conclusions

The conclusions, in fact, duplicate the annotation. This section should be revised. In view of the fact that the discussion of the results is divided into a large number of sections, it is necessary to summarize the results in the conclusion section. How does the ratio of monomers affect one or another parameter, what worsens, what improves? What, in the end, is the optimal composition of the composition?

It is necessary to indicate the prospects for further research of the obtained flame retardant, to suggest directions for the use of coated FNF.

The conclusion was revise as “in this paper, a novel P/N flame-retardant monomer (PDHAA) is successfully synthesized. The coating mixture is prepared by mixing PDHAA monomer and PM-2 monomer at different mass ratios, and then the coating mixture is applied to the surface of FNFs, obtaining coated FNFs with flame retardancy. With the increase of PDHAA monomer content, the LOI value of the sample shows an upward trend. The LOI value of FNFs-5 reaches 28.8%, which can self-extinguish rapidly in horizontal combustion test and pass the UL-94 V-0 test. At the same time, the coating changes into a smooth and dense carbon layer and releases non-flammable gases such as CO2 and NH3 during combustion. The formation of the carbon layer can reduce the heat transfer between the flame and the substrate, greatly reduce the emissions of CO and CO2. The FNFs-5 sample has the best flame retardancy. In addition, the introduction of PDHAA improves the mechanical properties of the coating.When PM-2 and PDHAA are blended together with a PM-2/PDHAA weight ratio of 2/1. Compared with FNFs, the tensile strength of FNFs-2 is improved by 58%. However, with the further increase of PDHAA content, the adhesion between coating and FNFs is deteriorates, and FNFs-1 has the best adhesion. In summary, FNFs with PDHAA monomers have higher flame retardant efficiency, it is of great significance to develop UV curable coatings containing multiple flame retardant elements. This coated FNFs has a wide application prospect in the field of fire protection of car interior decoration”.

Round 2

Reviewer 1 Report

It can be acceptted.

Author Response

Thank you for your recognition of our work

Reviewer 3 Report

There is a typo in the text. Instead of trimethylammonium hydrochloride, triethylammonium hydrochloride should be indicated.

The rest of the comments have been fixed.

Author Response

Page 2 ,Section2,''trimethylammonium hydrochloride'' was revise as ''triethylammonium hydrochloride''.
